# Effect of Heating under Different Vacuum Levels on Physicochemical and Oxidative Properties of Beef Sirloin

**DOI:** 10.3390/foods12071393

**Published:** 2023-03-24

**Authors:** Ah-Na Kim, Kyo-Yeon Lee, Chae Eun Park, Sung-Gil Choi

**Affiliations:** 1Digital Factory Project Group, Korea Food Research Institute, Wanju-gun 55365, Republic of Korea; 2Department of Food Science and Technology, Institute of Agriculture and Life Sciences, Gyeongsang National University, Jinju 52828, Republic of Korea; 3Division of Applied Life Science (BK21 Plus), Gyeongsang National University, Jinju 52828, Republic of Korea

**Keywords:** oxygen, vacuum, heating, cooking, beef sirloin, meat color, lipid oxidation

## Abstract

The physicochemical and oxidative properties of beef sirloin slices heated under atmospheric (101.33 kPa, a vacuum percent of 0%, control) and vacuum (50.8 kPa, 50% and 7.2 Pa, 99.99%) conditions by using an airtight vacuum container were compared. Heating at a higher vacuum level resulted in the lowest pH and cooking loss compared with the other conditions (*p* < 0.05). The beef in vacuum groups was less hard, chewy, and gummy than the control group, without any significant differences between the vacuum groups. More structural shrinkage and lower browning were observed in the meat heated at higher vacuum levels. Similarly, higher vacuum levels suppressed increases in the lightness (*L**), redness (*a**), and total color difference (*E**) of the surface after heating. The thiobarbituric acid (TBA) values, a sensitive indicator of meat oxidation due to heating, were only influenced by the vacuum conditions. Consequently, applying a vacuum effectively prevents the degradation in the meat’s physicochemical and oxidative properties during heating. The findings are useful for the sous-vide industry because they scientifically demonstrate how vacuum pressure affects the physicochemical and oxidative properties of the meat by using a specially designed airtight vacuum container.

## 1. Introduction

Meat and meat-based products are essential components of our diet that are mainly cooked before eating. Cooking makes meat more palatable and safer and hence is critical to the meat industry. However, improper heating can cause undesirable changes such as loss of nutritive value mainly due to lipid oxidation and degradation of heat-labile substances [1,2]. Thermal treatment also causes several physical changes, including texture modification [3]. Thus, the cooking methods and conditions are essential factors that affect the final quality of meat products [3,4].

Understanding the effect of cooking methods and conditions is necessary to enhance the edibility of meat. However, the meat industry is challenged by consumers’ perception of nutrition [5]. Notably, exposure to oxygen causes lipid oxidation and oxidative deterioration of meat products [6], which compromises its sensory value by producing undesirable odors, rancidity, decreased tenderness, loss of essential fatty acids, and toxic compounds [7,8]. Therefore, heat treatment technology has evolved to reduce the undesirable changes caused due to oxidation of cooked meats. Prominently, the “sous-vide” method improves meat texture and taste, resulting in lower nutrient losses and generation of hazardous compounds during cooking compared to traditional cooking methods [5]. This method, also called the “cook-in-bag” technique, differs from traditional methods, as the raw material is vacuum-sealed and then cooked [9]. Generally, vacuum packaging creates an oxygen-free environment, which effectively improves oxidative stability, maintains the quality, and/or extends the shelf-life [10]. Vacuum sealing prevents the development of off flavors due to oxidation and evaporative losses of flavor volatiles and moisture during heating [9]. However, as the meats are roughly packed under low (60–80%) vacuum conditions during the sous-vide process, the residual oxygen in the packs can affect the meat quality [11,12]. Many studies have reported that the residual oxygen in packs has been shown to be an important determinant of the quality of a meat product and can be correlated with quality parameters, especially with respect to color and texture, reflecting the final food quality [11,12,13]. Therefore, a systematic study of the effect of vacuum levels during heating on meat quality is crucial and has not been previously explored.

In this study, sirloin beef, which is one of representative beefsteak, was used. We assessed the effect of different vacuum levels during heating on the physicochemical properties such as pH, cooking loss, texture, color, and oxidative properties such as 2-thiobarbituric acid (TBA), acid, peroxide values, and fatty acid composition of the sirloin beef slices. The sirloin beef was packed using a specially designed airtight vacuum container which is able to completely isolate it from external environment and then heated at two vacuum levels such as the half of vacuum level (50.8 kPa of vacuum level and 50% of vacuum percent) and extremely high vacuum level (7.2 Pa and 99.99%) to determine the exact cause of the vacuum effect during heating. Accordingly, we demonstrated how the lack of oxygen during heating significantly affects the overall properties of the meat.

## 2. Materials and Methods

### 2.1. Meat Samples

We used the sirloin cuts of South Korean Hanwoo beef (Hapcheon-gun, Republic of Korea), which were around 2 years old in this study. Three carcasses were selected and slaughtered on the same day using the standard routines of a commercial slaughter plant in July 2021. The sirloin muscles were excised, vacuum packed, and aged in a cold room for 7 days at 2 °C. Four slices of each matured meat, weighing approximately 20 g, were cut (4 × 4 × 1 cm^3^, length × width × height), packed, and heated under four different vacuum levels.

### 2.2. Chemicals and Reagents

All the chemicals, including butylated hydroxyanisole (BHA), 2-thiobarbituric acid (TBA), trichloroacetic acid (TCA), malondialdehyde (MDA), 14% boron trifluoride-methanol solution (BF3), nonadecane, and solvents such as chloroform and methanol were obtained from Sigma-Aldrich (St. Louis, MO, USA). They were of analytical or LC/MS grade.

### 2.3. Packaging and Heating at Different Vacuum Levels

We used an airtight vacuum container, which we previously designed [14,15,16,17], to completely seal the meat to maintain the desired vacuum pressure during processing (Figure 1a,b). Three slices collected from three carcasses were packed separately in containers for each vacuum condition. The internal vacuum level of the container was adjusted using a vacuum pump (Figure 1a6). The valve (Figure 1a4) was closed when the desired vacuum level (V) was reached. The vacuum levels were set at 7.2 Pa (V99; vacuum level: 99.99%) or 50.8 kPa (V50; 50%) and were double-checked using a digital vacuum gauge (VGW-760K, Elitech Technology, Inc., Republic of Korea). The meat packaged at atmospheric pressure (101.33 kPa) was the control (C group; 0%).

The packed samples were immediately cooked at 95 °C for 30 min using a water bath (Figure 1c) and cooled with cold water for 30 min. During cooking, the core temperature was maintained at 75 °C using a copper-constantan thermocouple for all sample groups. The changes in pH, cooking loss, texture, color, TBA, acid, peroxide values, and fatty acid composition of the raw and cooked slices were investigated and compared.

### 2.4. pH

The raw and cooked sirloin slices (5 g) were minced and homogenized with 20 mL of distilled water for 1 min at 20,000 rpm with a D-500 homogenizer (Wiggen Hauser, Berlin, Germany). The pH of the mixtures was measured using an Istek Model 735P pH meter (Istek, Seoul, Republic of Korea) at room temperature.

### 2.5. Cooking Loss

The cooking loss of the sirloin slices was determined using the method described by Marcinkowska-Lesiak et al. [18] and estimated as the percentage of the weight of the cooked samples compared to the weight of the raw samples.

### 2.6. Texture

The texture of the raw and cooked sirloin slices was analyzed using a texture analyzer (Model TA-XT2; Texture Technologies Corp., Scarsdale, NY, USA) equipped with a 5 kg load cell and a TA-40 probe (10 cm diameter). Texture profile analysis (TPA) parameters such as hardness, springiness, chewiness, gumminess, and cohesiveness were measured at 1 mm/s of pre-test, test, and return speed at room temperature.

### 2.7. Color Measurement

The surface color of the raw and cooked sirloin slices was determined using a reflectance colorimeter (Chroma meter CR–400; Konica Minolta Sensing Inc., Osaka, Japan) at room temperature. It was measured at five randomly selected locations for each sample and recorded as color coordinates such as lightness (*L** ± dark-light), redness (*a** ± red-green), and yellowness (*b** ± yellow-blue). A numerical total color difference (*E**) between the raw and cooked meats was calculated by Equation (1).
(1)$$\Delta E^{*}={\left[{\left(L_{2}{}^{*}-L_{1}{}^{*}\right)}^{2}+{\left(a_{2}{}^{*}-a_{1}{}^{*}\right)}^{2}+{\left(b_{2}{}^{*}-b_{1}{}^{*}\right)}^{2}\right]}^{1/2}$$

### 2.8. TBA Value

Lipid oxidation, indicated by the TBA value, of the raw and cooked sirloin slices was determined using a method by Zhu et al. [19]. Briefly, 5 g of the sample was weighed in a 50 mL test tube and homogenized with 15 mL of distilled water and 50 μL of 7.2% BHA at 10,000 rpm using the D-500 homogenizer. The mixture was filtered using Whatman No.1 filter paper (Maidstone, UK). Next, 1 mL of the filtrate and 2 mL of TBA reagent consisting of TBA/TCA (15 mM/15%) were added into a 15 mL test tube and then immersed in a boiling water bath for 1 min to develop color. The reactant was cooled for 5 min under tap water and centrifuged for 15 min at 2500× *g* at 4 °C. The absorbance was measured at 531 nm using a UV-vis spectrophotometer (UV-1800, Shimadzu corporation, Kyoto, Japan) against a blank containing 1 mL distilled water and 2 mL TBA reagent. The TBA value was expressed as mg of malondialdehyde (MDA) per kg sample according to a standard curve for MDA.

### 2.9. Acid Value

The acid values of the raw and cooked sirloin slices were determined according to the AOCS official method Cd 3d-63 [20] with modifications. Lipids were extracted by homogenizing the sample with 1:2 ethanol-ether (*v*/*v*) at 10,000 rpm for 1 min. Results were expressed as mg potassium hydroxide (KOH) per g meat sample (mg KOH/g).

### 2.10. Peroxide Value

The peroxide values of the raw and cooked sirloin slices were determined according to Pegg [21] with minor modifications. After homogenizing the sample (3 g), lipids were extracted with chloroform (30 mL) at 10,000 rpm for 1 min. The results were expressed as milliequivalents (meq) per kg meat sample (meq/kg).

### 2.11. Fatty Acids Analysis

The fatty acid content of the raw and cooked sirloin slices was determined using a method by Folch et al. [22] with minor modifications. The sample (2.5 g) was homogenized with 10 mL of Folch solution (chloroform:methanol, 2:1, *v*/*v*) at 10,000 rpm for 30 s using the D-500 homogenizer. The homogenate was filtered with Whatman No.1 filter paper (Maidstone, UK). Then, 2.5 mL of 0.88% NaCl was added to the filtrate, vortexed for 30 s, and allowed to stand for 1 h. For saponification, 0.5 M NaOH was added to 1 mL of the bottom layer (chloroform–fat fraction) and hydrolyzed for 5 min at 90 °C. After cooling to room temperature for 5 min, it was methylated at 90 °C for 5 min after adding 14% BF3/methanol solution (1 mL) and then cooled for 5 min. Next, 2 mL each of hexane and saturated NaCl was added to the reactant, vortexed for 30 s, and allowed to stand for 30 min. The hexane layer was collected and stored in a −80 °C freezer until analysis.

Fatty acid methyl esters were separated using a Gas Chromatograph Mass Spectrometer Triple Quadrupole (GCMS-TQ8050, Shimadzu, Kyoto, Japan) equipped with a Supelco SP-2560 GC column (100 m × 0.25 mm internal diameter × 0.2 μm film thickness) and a Flame Ionization Detector. An amount of 2 μL of the collected sample was introduced to the split injection port. The oven temperature was maintained at 100 °C for 4 min, ramped up to 240 °C at a rate of 3 °C/min, and then held for 15 min. The injector and detector temperatures were maintained at 225 and 285 °C, respectively. C19:0 was used as an internal standard. The analytes were identified using GC/MS databases. The fatty acid compositions of the raw and cooked sirloin slices were expressed as the proportion of total fatty acid methyl esters.

### 2.12. Statistical Analysis

Data are reported as means ± standard deviation of three independent experiments. Statistical analyses were performed using analysis of variance with the SAS statistical software (v 9.4 version, SAS Institute Inc., Cary, NC, USA). Significant differences (*p* < 0.05) among the groups were determined using Duncan’s multiple range tests.

## 3. Results and Discussion

### 3.1. pH and Cooking Loss

The pH and cooking loss of the sirloin beef slices cooked at different vacuum levels are shown in Table 1. Heating significantly affected the pH of all groups. After heating, the pH, which was 5.57 in the raw sample, increased to 5.86, 5.85, and 5.76 in the C, V50, and V99 groups, respectively. Similarly, Oz and Zikirov [23] reported that the pH of beef chops increased from 5.56 to between 5.73 and 5.94 after cooking. This rise in pH is mainly due to the cleavage of bonds in the imidazole, sulfhydryl, and hydroxyl groups [24]. In particular, the pH levels of the C and V50 groups were the highest without any significant mutual difference. Contrastingly, the V99 group (with the highest vacuum level) showed the lowest pH increase rate after heating.

The cooking loss results were consistent with those of pH. There was no significant difference in cooking loss between the C and V50 groups, while the V99 group, with the highest vacuum level, displayed a 7.66% decrease compared to the C group, which showed the lowest cooking loss. Cooking loss occurs due to heat-induced protein denaturation and shrinkage, which releases a combination of liquid and soluble matter [25]. The cooking method considerably influences the cooking loss of meat [26]. Jeong et al. [27] and Hwang et al. [28] reported that sous-vide cooking (cooking under a vacuum) results in a significantly lower cooking loss compared to meat heated without a vacuum. In this technique, the water loss from the meat is prevented because of the vacuum packing, minimizing shrinkage and juice loss [29]. Furthermore, the free acidic groups released during cooking might increase the pH of the meat [30]. The results indicated that vacuum packaging at 7.2 Pa inhibits cooking loss and raises the pH.

### 3.2. Texture

A textural profile analysis (TPA) is used to obtain complementary and detailed information about the textural characteristics and to predict the texture of cooked meat [31]. The texture of cooked meat is a very important factor for consumers, and hardness, springiness, chewiness, gumminess, and cohesiveness are all key attributes that contribute to the overall sensory experience of eating cooked meat. Texture parameters such as hardness, springiness, chewiness, gumminess, and cohesiveness refer to a measure of its tenderness, how well the meat recovers its shape after being deformed, the amount of work required to chew the meat, the degree of adhesiveness of the meat, and how well the meat holds together when chewed, respectively [32]. The parameters of sirloin beef slices cooked at different vacuum levels are presented in Table 2. Initially, we observed similar trends in these parameters. The cooking process significantly increased the hardness, chewiness, and gumminess of the sirloin beef slices compared to raw meat. The highest parameters were observed in the C group (meat cooked under atmosphere conditions). Generally, muscle meat undergoes substantial structural changes during heating. The muscle fibers shrink transversely and longitudinally, the sarcoplasmic proteins aggregate and gel, and connective tissue shorten and solubilize [9,33]. Heat dissolves collagen, thereby increasing the tenderness. However, it also denatures myofibrillar protein, which makes the meat tougher [34]. While the meat in the vacuum groups, including V50 and V99, exhibited a lower increase compared to the C group, there was no significant difference between them. Consistent with this, heating under vacuum conditions such as sous-vide cooking decreases the hardness, chewiness, and gumminess of the beef steaks compared with those heated under atmospheric conditions [35]. Reports have also shown that the meat products cooked by sous-vide were less hard than the conventionally cooked ones [36,37,38]. A micrographic study by Garcia-Segovia et al. [33] proved that beef muscle cooked under atmospheric pressure was harder and chewier than that cooked sous-vide. This might be because sous-vide cooking produces more diffused connective tissue (endomysium) structures, less compact myofiber-sarcoplasms, greater collagen solvation, and lower gel aggregate formation. Rinaldi et al. [38] reported that sous-vide cooking dissolves connective tissues, making meat tender and reducing its hardness. Therefore, cooking under a vacuum limits the reduction in tenderness and juiciness and increases the toughness, resulting in a higher palatability than conventional cooking [39]. The hardness and chewiness of cooked meat were negatively correlated with tenderness, juiciness, and sensory quality [40]. Additionally, cooking losses directly contribute to meat toughness because it reflects the shrinkage of the longitudinal muscle fibers during heating [1]. The appearance of the sirloin beef surfaces heated at different vacuum levels validated our observations (Figure 2), showing considerable shrinkage of meat in the C group (heated under atmospheric conditions) compared to the V groups. Cooking under a vacuum effectively improves the texture by suppressing the increase in hardness, chewiness, and gumminess of the sirloin beef slices. Applying a vacuum at 50 kPa might be enough to inhibit the structural changes induced by heat as we did not observe any significant difference between cooking at 7.2 Pa (highest vacuum level) and 50.8 kPa. Unlike the observed texture parameters, heating under a vacuum decreased the springiness without any significant difference between different vacuum levels. This indicated that the meats in all the groups had similar elasticity and could return to their original length after stretching. Similarly, Rinaldi et al. [38] reported that springiness did not significantly differ among beef samples cooked by conventional and sous-vide methods. In terms of cohesiveness, we did not observe a meaningful trend, even though the V99 group showed the highest level among all the groups.

### 3.3. Color and Appearance

We observed significant color changes on the surface and cross-sections of the sirloin beef slices (Figure 2). The surface of the meat cooked under a higher vacuum level showed less browning and was redder. Mainly, the surface color of the meat in V99 (cooked under deficient oxygen levels) was similar to that of the raw meat, showing minimal browning. The centers of the internal cross-section areas of all cooked groups appeared pink. Notably, a brown-colored band was noticed on the outer side of the C samples, while the V50 samples were more evenly browned. Conversely, we did not observe browning in the V99 sample, which was consistent with the surface color.

The obtained chromatic parameters supported these findings. Table 3 shows the color parameters of the surface of the beef slice heated under different vacuum levels, including lightness (*L**), redness (*a**), and yellowness (*b**). No meaningful trend was seen for yellowness (*b**) based on the heating and vacuum levels. However, the vacuum level during heating significantly affected *L** and *a**. First, the heat processing increased *L** and decreased *a** for all groups. Among all cooked groups, the V99 group had the lowest *L**. It also had the highest *a**, significantly higher than C and V50 groups. Due the significant differences in the *L** and *a** values of the treated groups, the total color of the meat heated under a higher vacuum level was similar to that of raw meat. Meat turns brown during cooking due to myoglobin (Mb) denaturation and Maillard reactions [40]. Mb is the primary protein responsible for meat color. Depending on its redox state, it can impart pinkish-purple, bright-red, or brown colors, corresponding to deoxymyoglobin (DMb), oxymyoglobin (OMb), and metmyoglobin (MMb), respectively [41]. OMb is deoxygenated to an unstable form, DMb, under low oxygen levels. The ferrous Mbs, namely OMb and DMb, are mainly oxidized to ferric Mb or MMb by enzymes and spoilage bacteria during storage rather than heating [42]. MMb forms the brown globin hemichromogen or ferrihemochrome when the globin is heat-denatured. OMb and MMb are denatured to the red globin hemochromogen, also known as ferrohemochrome, which is then readily oxidized to brown ferrohemochrome, the pigment that imparts a dull-brown color to cooked meat [41]. Therefore, protein denaturation increases during heating, resulting in higher brightness and lower redness of meat products [42]. However, in this study, the absence of oxidation due to the lack of oxygen reduced the formation of OMb and DMb and reduced the extensive Maillard reactions, resulting in the least browning in the V99 group [43]. The V50 group also might be influenced by decreased oxidation and Maillard reactivity due to low oxygen levels during heating, resulting in a higher *a** than the C group. Hunt et al. [41] showed that the meat cooked under a vacuum is redder because of Mb changes caused by oxygen unavailability, with DMb being the most resistant to protein denaturation. Dominguez-Hernandes et al. [4] reported that the significant reddish-pink color of sous-vide cooked meat is because of OMb or DMb, while sarcoplasmic proteins also deposit inside fibers, forming gel and giving the meat a swollen and compact texture.

### 3.4. TBA Value, Acid Value, and Peroxide Value

Lipid oxidation of meat products is commonly determined using TBA, acid, and peroxide values [44]. We evaluated the lipid oxidation of the sirloin beef slices heated under different vacuum levels (Table 1) using these three parameters and observed trends based on the vacuum levels. The TBA value of the C group only increased after cooking, while those of the V groups were similar to that of raw meat before heating. Moreover, the acid values of all cooked groups increased after cooking without any significant differences between them. Contrastingly, heating under different vacuum levels did not affect the beef’s peroxide value.

Lipids are mainly oxidized in three ways: autoxidation, enzyme-catalyzed oxidation, and photo-oxidation [45]. Among them, lipid autoxidation is the predominant mode of lipid oxidation in meat products, consisting of a primary and a secondary phase. The primary phase is initiated by interaction with oxygen in the presence of initiators such as heat, light, and high-energy radiation. Lipid hydroperoxides affect the peroxide value, which indicates the formation of primary oxidation compounds in meat. However, the hydroperoxides are unstable and decompose rapidly, and their content decreases as oxidation increases. The secondary oxidation phase involves the formation of oxidation products such as dialdehydes with three carbons, including MDA. It can be measured using TBA values as a lipid oxidation marker. The secondary products are more stable and reflect the consequences of oxidation, such as a deterioration in quality and appearance and the development of rancid flavor and odors. Guillen-Sans and Guzman-Chozas [46] reported that the TBA value is a more sensitive indicator than others, such as the peroxide value, and shows a high correlation with the oxidized products in animal foods. Additionally, the decomposition of lipids produces fatty acids in meats, indicating hydrolytic rancidity. Other types of lipid oxidation can also produce acids. The acid value is useful to determine the free fatty acid composition of a sample, reflecting their formation or decomposition [47]. Heating results in the formation of free fatty acids due to a preferential release of the shorter-chain and unsaturated fatty acids by thermal hydrolysis of triglycerides [48]. The acid value and peroxide value are a measure of the early stages and the initial stages of lipid oxidation, respectively. Here, we showed that the vacuum level and heat might affect the acid values. Heat treatment disrupts the membranes’ integrity and exposes the phospholipids to molecular oxygen, oxidative enzymes, heme pigments, and metal ions in meat [49]. Several reports have shown that lipid oxidation is accelerated by heating, and hence cooked meat gets oxidized faster than raw meat [8,46]. We showed that the low oxygen levels due to vacuum prevented lipid autooxidation, although it could not limit lipid hydrolysis during heating. In other words, the vacuum condition can prevent secondary lipid oxidation, although there is no effect on the early and initial stages of lipid oxidation. This suggests that the TBA value is the best indicator of the degree of lipid rancidity of the beef heated under different vacuum conditions. The TBA assay results indicated that the secondary lipid oxidation of the V groups did not occur, while secondary products were produced in the C group. When lipid oxidation is initiated, oxygen is activated via an energy source such as temperature [50,51]. However, it is difficult for unsaturated fatty acids to react with active oxygen species under low oxygen level (vacuum levels 50 kPa) [47]. Furthermore, the lag phase of the initiation stage might limit the oxidation process during vacuum heating because the free radicals preferentially oxidize the natural antioxidants in meat, protecting the fatty acids during the early oxidation phase [52].

### 3.5. Fatty Acid Composition

Qualitative measurements of fatty acids in raw meat and heated groups were conducted by GC/MS and are presented in Table 4. The most abundant fatty acid was oleic acid (C18:1 n-9), followed by heptadecanoic acid (C17:0), palmitoleic acid (C16:1), linoleic acid (C18:2, n-6), tridecanoic acid (C13:0), and myristoleic acid (C14:1), as shown in previous studies [53,54]. The heating and different vacuum levels during cooking did not have any significant effect on the fatty acid compositions of the beef (*p* > 0.05), even though the TBA and acid values were altered. Namely, there were no changes in all types of fatty acids, including saturated (palmitic and stearic), monounsaturated (oleic), and polyunsaturated (linoleic and linolenic acid). The fatty acid composition is an important parameter that influences lipid oxidation as they are the oxidation substrates in the meat [4]. Studies have also reported lack of significant differences in the fatty acid compositions between raw and heated meat products [55,56]. The lack of correlation between the total fatty acid composition and heat process might be due to the low fat content in the meats and variability in the compositions depending on each sample [53,57,58]. Consequently, heating under vacuum limited lipid oxidation, which is more clearly indicated by the TBA value than the fatty acid composition.

## 4. Conclusions

The heating process increased the pH, cooking loss, and texture parameters, including hardness, chewiness, and gumminess. The V99 group showed the lowest increases in pH and cooking loss. Collectively, the V groups showed a lower increase in the textural parameters than the C group, without any significant difference between them. The difference in appearance supports the physicochemical analysis, as more structural denaturation, such as shrinkage and browning, were observed in the C group than in the V groups. A higher vacuum level resulted in lower browning and also resulted in lower *L**, *a**, and *E** values of the surface after heating. The color values mostly depended on the vacuum levels during heating. Particularly, there was no browning on the surface and cross-section area of the meat heated under insufficient oxygen levels (V99 group). There was no significant effect of heating under different vacuum levels on the oxidative properties, including acid, peroxide, and fatty acid composition values. However, the TBA value, a secondary lipid autooxidation indicator, was the most appropriate for evaluating heat-induced meat oxidation. The TBA values showed that the V groups were less oxidized than the C group, without any significant difference between V50 and V99 groups. In summary, this study demonstrated that heating under vacuum conditions effectively prevents changes in physical qualities and oxidative processes of the sirloin beef. Moreover, the limited oxygen at a vacuum level of 50 kPa is not sufficient to cause textural and oxidative degradation during cooking. These results elucidate the changes in physicochemical and oxidative properties based on the available oxygen levels during the cooking of meat. Furthermore, our findings can be used to establish the vacuum conditions while packaging meat for sous-vide cooking.

## Figures and Tables

**Figure 1 foods-12-01393-f001:**
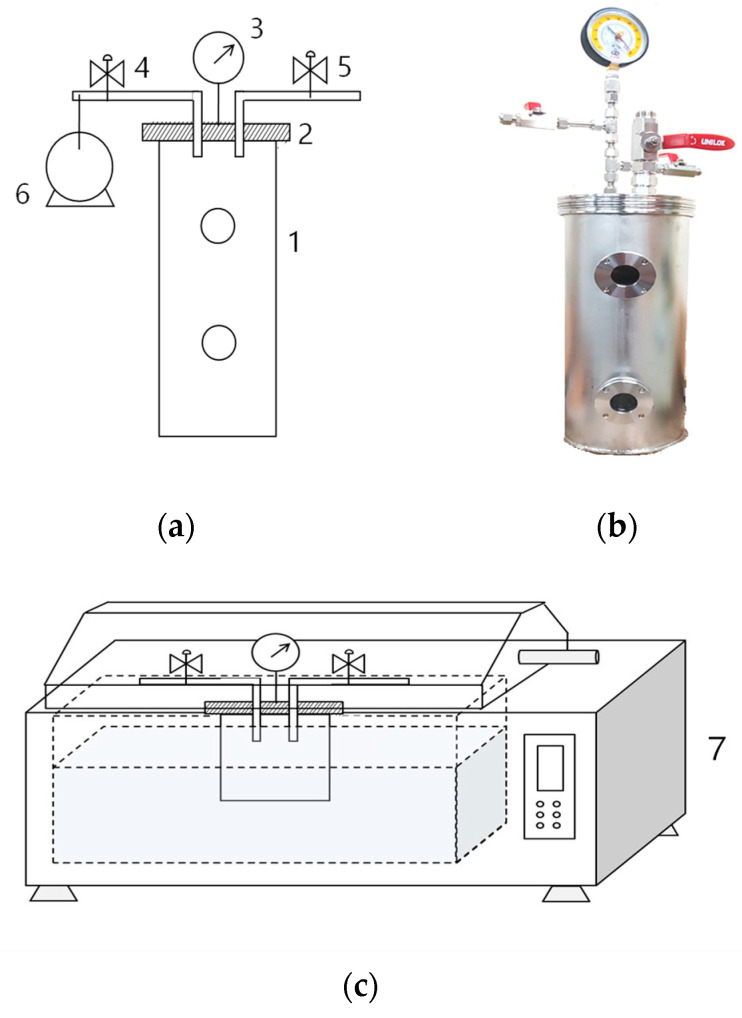
(**a**) A schematic diagram and (**b**) picture of the airtight vacuum container designed for maintaining vacuum level. (**c**) The design of the apparatus used for heat treatment of packaged sirloin beef slices: (1) sealable chamber, (2) airtight detachable lid, (3) vacuum gauge, (4 and 5) valves, (6) vacuum pump, and (7) water bath.

**Figure 2 foods-12-01393-f002:**
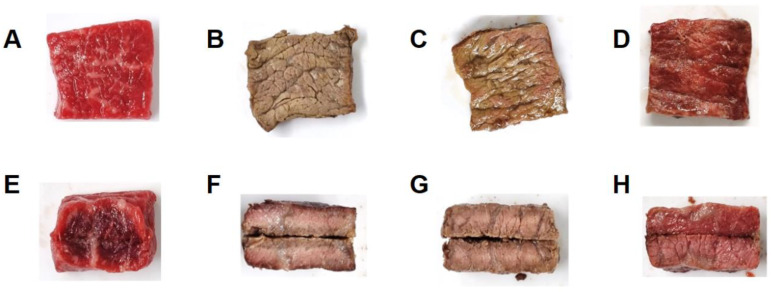
Appearance of raw sirloin beef slices (**A**,**E**) and the slices heated at different vacuum levels: (**B**,**F**) C group (vacuum percent and level; 0%, 101.33 kPa), (**C**,**G**) V50 group (50%, 50.8 kPa), and (**D**,**H**) V99 group (99.99%, 7.2 Pa). The top and bottom panels show the surface and cross-sections, respectively.

**Table 1 foods-12-01393-t001:** pH, cooking loss, and TBA, acid, and peroxide values of raw sirloin beef slices and the slices heated at different vacuum levels, i.e., the C group (vacuum percent and level; 0%, 101.33 kPa), V50 group (50.00%, 50 kPa), and V99 group (99.99%, 7.2 Pa).

Properties	Before Cooking	After Cooking		
		C	V50	V99
pH	5.57 ± 0.05 ^c^	5.86 ± 0.04 ^a^	5.85 ± 0.02 ^a^	5.76 ± 0.04 ^b^
Cooking loss (%)	-	27.27 ± 5.65 ^a^	26.61 ± 1.80 ^a^	19.61 ± 2.17 ^b^
TBA (mg MDA/kg)	6.28 ± 1.53 ^b^	7.44 ± 0.38 ^a^	6.30 ± 1.16 ^b^	6.35 ± 0.67 ^b^
Acid value (mg KOH/g)	0.15 ± 0.05 ^b^	0.54 ± 0.05 ^a^	0.47 ± 0.04 ^a^	0.53 ± 0.08 ^a^
Peroxide value (meq/kg)	4.85 ± 2.97 ^a^	4.87 ± 3.51 ^a^	4.86 ± 2.78 ^a^	4.82 ± 3.52 ^a^

Values marked by different letters within the same row indicate significant difference (*p* < 0.05). Abbreviations: TBA, thiobarbituric acid; KOH, potassium hydroxide.

**Table 2 foods-12-01393-t002:** The texture parameters of raw sirloin beef slices and the slices heated at different vacuum levels, including C (vacuum percent and level; 0%, 101.33 kPa), V50 (50%, 50 kPa), and V99 (99.99%, 7.2 Pa) groups.

Texture Parameters	Before Cooking	After Cooking		
		C	V50	V99
Hardness (g)	172.46 ± 39.47 ^c^	2593.10 ± 532.73 ^a^	1723.54 ± 284.57 ^b^	1561.12 ± 640.94 ^b^
Springiness	2.07 ± 0.02 ^a^	2.00 ± 0.01 ^b^	1.99 ± 0.03 ^b^	1.98 ± 0.02 ^b^
Chewiness	429.92 ± 127.18 ^c^	6459.53 ± 1246.10 ^a^	4063.27 ± 266.65 ^b^	5047.62 ± 719.74 ^b^
Gumminess	207.88 ± 62.87 ^c^	3229.32 ± 622.57 ^a^	1933.75 ± 270.79 ^b^	2159.31 ± 930.31 ^b^
Cohesiveness	1.19 ± 0.13 ^ab^	1.26 ± 0.22 ^ab^	1.13 ± 0.05 ^b^	1.38 ± 0.13 ^a^

Values marked by different letters within the same row indicate significant difference (*p* < 0.05).

**Table 3 foods-12-01393-t003:** The color values of sirloin beef slices heated at different vacuum levels, including C (vacuum percent and level; 0%, 101.33 kPa), V50 (50%, 50 kPa), and V99 (99.99%, 7.2 Pa) groups compared with raw beef.

Color Values	Before Cooking	After Cooking		
		C	V50	V99
*L**	43.06 ± 2.95 ^c^	50.05 ± 1.85 ^a^	49.10 ± 1.48 ^a^	44.82 ± 2.85 ^b^
*a**	24.40 ± 1.55 ^a^	6.04 ± 0.74 ^d^	7.33 ± 1.15 ^c^	17.39 ± 1.87 ^b^
*b**	11.83 ± 0.71 ^b^	10.06 ± 0.59 ^c^	12.69 ± 1.15 ^a^	9.80 ± 0.80 ^c^
*E**	-	19.81 ± 0.79 ^a^	18.21 ± 1.33 ^b^	7.99 ± 2.12 ^c^

Values marked by different letters within the same row indicate a significant difference (*p* < 0.05).

**Table 4 foods-12-01393-t004:** Fatty acid compositions of raw sirloin beef slices and the slices heated at different vacuum levels, including C (vacuum percent and level; 0%, 101.33 kPa), V50 (50%, 50 kPa), and V99 (99.99%, 7.2 Pa) groups. (Unit: %).

Fatty Acids	Before Cooking	After Cooking		
		C	V50	V99
C13:0	1.43 ± 0.13	1.39 ± 0.08	1.58 ± 0.12	1.57 ± 0.04
C14:1	0.56 ± 0.48	0.45 ± 0.39	0.96 ± 0.22	0.84 ± 0.05
C17:0	21.04 ± 0.26	20.53 ± 0.34	20.11 ± 0.13	20.78 ± 0.95
C16:1	3.83 ± 0.10	3.43 ± 0.46	4.27 ± 0.66	3.91 ± 0.24
C18:1(n-9)	71.48 ± 0.54	72.98 ± 1.51	71.82 ± 1.18	71.41 ± 1.06
C18:2(n-6)	1.64 ± 0.32	1.22 ± 0.60	1.26 ± 0.23	1.49 ± 0.15

## Data Availability

Data are contained within the article.

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
