# Peer review of "Effect of Heating under Different Vacuum Levels on Physicochemical and Oxidative Properties of Beef Sirloin"

_foods, 2023, doi:10.3390/foods12071393_

Round 1

Reviewer 1 Report

This study determine the effect of heating under different vacuum levels on physicochemical and oxidative properties of beef sirloin. The topic is interesting. If the experimental designs and discussion is well-structured, the findings will useful for sous-vide industry. However, in my opinion, this MS still required major revision. I also give comments as follow:

Major points:

- The introduction should add more information as suggest.

- Author should defense why determine only 2 level of vacuum and why are 50% and 99.9% since the 50% is too least while the 99.9% is difficult to use in the real situation/industry.

- In-depth exploration/discussion should provide. (not too general) This study should discuss related to vacuum level, not the changes after heating.

Here is my comments in details:

ABSTRACT

- Line13: The percent of vacuum level should provide (50 and 99.99%), together with 50.8 and 7.2 Pa.

- Line14: I suggested to change into “higher” since there are only 2 conditions of vacuum.  

- Line14: Please change into “the higher vacuum level contained the lowest pH and cooking loss, compared with others (P<0.05)” (At this point please check the results, is it significant different or not?)

- Line17: Higher temperature??? Is this study also varied the temperature for cooking? If yes, author should scope/state in the first sentence as well.

- I suggested adding some empirical results, particularly quantitative data to strengthen the abstract. Moreover, the weak point of this MS is lacking new findings/novel. Author should clearly provide the new findings in the abstract or give ideas for further use/further studies/further application. (Key point: This study is useful for the sous-vide industry, thus, it should be appeared in abstract to drawn people attention).

INTRODUCTION

- Line 46: It is interesting. Normally, meats were packed under >80% of vacuum for sous-vide cooking. There are many previous research works on those conditions. The information that “meats are roughly packed under low (60%–80%) vacuum conditions during the sous-vide process” should provide in details and referred to references. Author should review some emphasis results related to those conditions. And also, author should provide the hypothesis, which leads to set the vacuum level of 50.8 kPa and 7.2 Pa in this study.

- I suggested to improve the introduction. Some emphasis information should be added. For example, the reviews of vacuum levels should be included, effects of vacuum levels on physicochemical parameters or meat quality of previous studies should be stated or the effect of various oxygen amount on meat quality should provide, also, the specific characteristics of raw material (beef sirloin) should be review, etc.

MATERIAL AND METHODS

- Line67-68: the sample is too small since the real situation of sous-vide meat/beef is preferred to do with larger piece (approximately 150-300 g). What the reason to prepare samples like this? Please describe.

- Line76-77: Why author designed only these 2 conditions of vacuum level. I think 99.99% is difficult to do in the real industry and 50% is too least. How’s about 60, 70, 80%? Also, if have only 2 conditions, we cannot determine the trends of vacuum level, which impacted to the oxygen leaking during processing and meat quality.

- Line82: How’s changes of meat temperature during processing? Is it different or not between each others? This information should appear, for at least, in the complement data.

- Line107: Gumminess is referred to semi-solid sample. Why this parameter is reported since the meat is solid? Please describe.

RESULTS AND DISCUSSION

- Line173-174: The rise of pH that author discuss seem to the effect during prolong heating. It is not related to the vacuum level. Thus, please add the reason to support why V99 provide lower pH than others.

- For 3.2 Texture parameters. The discussion should be improved. Author should state and give a discussion why each parameter changed like that. How’s important of hardness, springiness, chewiness, gumminess, and cohesiveness (how’s it referred to meat quality), etc. Again, I think gumminess is for semi-solid sample, author classified the beef as semi-solid?

- Fig.2 Why some beef still red? It means that each sample did not well-cooked at the same level? If yes, the result may not cause by the vacuum level only?

- Please add the discussion related to delE*. What author used as standard for them. How’s this parameter referred to meat quality.

- TBA value, acid value, and peroxide value is all referred to lipid oxidation, but different states/step. This information should be add/discuss more.

- Table4: There was no significant difference in all FA competitions among samples, thus the letter “a” is no need.

CONCLUSION

- Conclusion should be shorten and provide only main results. I suggested to end up with the further application/study.

Author Response

Author's Reply to the Review

Dear Editor

We would like to thank you for your consideration and suggestions on our manuscript ‘foods-2233829’, entitled ‘Effect of heating under different vacuum levels on physicochemical and oxidative properties of beef sirloin’. We found the comments helpful and believe our revised the manuscript represents a significant improvement over our initial submission. We have reviewed the comments of the reviewers and have accordingly revised manuscript. In the following pages there are our point-by-point responses to each of the comments of the reviewers.

Reviewers’ comments

Reviewer  #1

This study determine the effect of heating under different vacuum levels on physicochemical and oxidative properties of beef sirloin. The topic is interesting. If the experimental designs and discussion is well-structured, the findings will useful for sous-vide industry. However, in my opinion, this MS still required major revision. I also give comments as follow:

Major points:

- The introduction should add more information as suggest.

- Author should defense why determine only 2 level of vacuum and why are 50% and 99.9% since the 50% is too least while the 99.9% is difficult to use in the real situation/industry.

- In-depth exploration/discussion should provide. (not too general) This study should discuss related to vacuum level, not the changes after heating.

Here is my comments in details:

 [1]

Comment: [ ABSTRACT ]

- Line13: The percent of vacuum level should provide (50 and 99.99%), together with 50.8 and 7.2 Pa.

- Line14: I suggested to change into “higher” since there are only 2 conditions of vacuum. 

- Line14: Please change into “the higher vacuum level contained the lowest pH and cooking loss, compared with others (P<0.05)” (At this point please check the results, is it significant different or not?)

- Line17: Higher temperature??? Is this study also varied the temperature for cooking? If yes, author should scope/state in the first sentence as well.

- I suggested adding some empirical results, particularly quantitative data to strengthen the abstract. Moreover, the weak point of this MS is lacking new findings/novel. Author should clearly provide the new findings in the abstract or give ideas for further use/further studies/further application. (Key point: This study is useful for the sous-vide industry, thus, it should be appeared in abstract to drawn people attention).

Response: As a reviewer comments, we have revised and improved the abstract (L13-25).

[2]

Comment: [ INTRODUCTION ]

- Line 46: It is interesting. Normally, meats were packed under >80% of vacuum for sous-vide cooking. There are many previous research works on those conditions. The information that “meats are roughly packed under low (60%–80%) vacuum conditions during the sous-vide process” should provide in details and referred to references. Author should review some emphasis results related to those conditions. And also, author should provide the hypothesis, which leads to set the vacuum level of 50.8 kPa and 7.2 Pa in this study.

- I suggested to improve the introduction. Some emphasis information should be added. For example, the reviews of vacuum levels should be included, effects of vacuum levels on physicochemical parameters or meat quality of previous studies should be stated or the effect of various oxygen amount on meat quality should provide, also, the specific characteristics of raw material (beef sirloin) should be review, etc.

Response: As a reviewer comments, we have added more information about the residual oxygen in packs and some emphasis information (L52-66). In addition, the objective of this study is to ascertain the effect of vacuum level including the half of the vacuum percent (V50) and an extremely high vacuum level (V99) during heating on the physicochemical and oxidative properties of sirloin beef using a specially designed airtight vacuum container which is completely isolated from the external environment, not vacuumized polypropylene pouch. We have supplemented the objective and design of this study.

[3]

Comment: [ MATERIAL AND METHODS ]

- Line67-68: the sample is too small since the real situation of sous-vide meat/beef is preferred to do with larger piece (approximately 150-300 g). What the reason to prepare samples like this? Please describe.

- Line76-77: Why author designed only these 2 conditions of vacuum level. I think 99.99% is difficult to do in the real industry and 50% is too least. How’s about 60, 70, 80%? Also, if have only 2 conditions, we cannot determine the trends of vacuum level, which impacted to the oxygen leaking during processing and meat quality.

- Line82: How’s changes of meat temperature during processing? Is it different or not between each others? This information should appear, for at least, in the complement data.

- Line107: Gumminess is referred to semi-solid sample. Why this parameter is reported since the meat is solid? Please describe.

Response: The sample size was limited due to the size of the vacuum container. Based on the scientific data of this study, a further study will be conducted according to various vacuum degrees and heat treatment temperatures with meat of the actual size used in meat industry. As mentioned above, the purpose of this study is to ascertain the effect of vacuum level including the half of the vacuum percent (V50) and an extremely high vacuum level (V99) during heating on the physicochemical and oxidative properties of sirloin beef using a specially designed airtight vacuum container which is completely isolated from the external environment, not vacuumized polypropylene pouch. In addition, there was no significant difference between V50 group and V99 group for the oxidative properties, including acid, peroxide, TBA, and fatty acid composition values and texture properties (hardness, springiness, chewiness, and gumminess). This finding indicated that the properties of the meat did not change more sensitively than expected depending on the vacuum level during the thermal treatment, while meat color was the most sensitive.

A temperature change cannot measured, but a core temperature was only measured after heating at 95°C for 30 min because of the use of an airtight vacuum container.

Gumminess is a texture attribute that describes how chewy and sticky a food feels in the mouth. Many studies are often used to describe the texture of meat, as well as other foods like gummy candies even though gumminess is referred to semi-solid sample [1-3]. Nevertheless, if you think it is not appropriate, we will revise it.

  1. Breene, W. M. (1975). Application of texture profile analysis to instrumental food texture evaluation. Journal of texture Studies, 6(1), 53-82
  2. Zhuang, X., Han, M., Kang, Z. L., Wang, K., Bai, Y., Xu, X. L., & Zhou, G. H. (2016). Effects of the sugarcane dietary fiber and pre-emulsified sesame oil on low-fat meat batter physicochemical property, texture, and microstructure. Meat science, 113, 107-115.
  3. Choe, J. H., Choi, M. H., Rhee, M. S., & Kim, B. C. (2016). Estimation of sensory pork loin tenderness using Warner-Bratzler shear force and texture profile analysis measurements. Asian-Australasian journal of animal sciences, 29(7), 1029.

[4]

Comment: [ RESULTS AND DISCUSSION ]

- Line173-174: The rise of pH that author discuss seem to the effect during prolong heating. It is not related to the vacuum level. Thus, please add the reason to support why V99 provide lower pH than others.

- For 3.2 Texture parameters. The discussion should be improved. Author should state and give a discussion why each parameter changed like that. How’s important of hardness, springiness, chewiness, gumminess, and cohesiveness (how’s it referred to meat quality), etc. Again, I think gumminess is for semi-solid sample, author classified the beef as semi-solid?

- Fig.2 Why some beef still red? It means that each sample did not well-cooked at the same level? If yes, the result may not cause by the vacuum level only?

- Please add the discussion related to delE*. What author used as standard for them. How’s this parameter referred to meat quality.

- TBA value, acid value, and peroxide value is all referred to lipid oxidation, but different states/step. This information should be add/discuss more.

- Table4: There was no significant difference in all FA competitions among samples, thus the letter “a” is no need.

Response: We have already discussed why V99 provide lower pH than others (L199-201). If you think need further discussion, please make a request.

In addition, we have improved the discussion of '3.2. Texture’ (L209-216 and L235-236).

In the context of meat quality during storage or display, delE* are often used to assess the freshness and shelf life of meat products. A higher delE* indicates that the meat has undergone more color change and may be less fresh or have a shorter shelf life. On the other hand, a lower delta E value indicates that the meat has undergone less color change and may be fresher and have a longer shelf life. In case of meat cooked under atmospheric condition, delE* is also used to measure color changes, particularly in relation to the degree of doneness. delE* is a numerical value that expresses the difference in color between two samples, and is calculated by comparing the color of a cooked meat sample with a reference sample of the same meat that has been cooked to a standard degree of doneness. On the other hand, this result indicated that vacuum groups had low delE* value, compared to C group. The difference in meat color are considered to be a cause of vacuum level since the core temperature was smoothly transferred, and the texture was similar to C groups.

In addition, we have added more discussion about lipid oxidation (L335-336 and L342-344).

[5]

Comment: [ CONCLUSION ]

- Conclusion should be shorten and provide only main results. I suggested to end up with the further application/study.

Response: As commented, we have shorten '4. Concolusion' (L376-380, 384, and 388-390).

In addition, we have added several references (L435-438 and 472-473).

We hope you will consider this paper as suitable for publication in your journal.

Best regards.

*Corresponding author: Sung-Gil Choi, Professor

Division of Food Science and Technology (Institute of Agriculture and Life Sciences), Gyeongsang National University, Jinju 52828, Korea

phone : +82-10-7143-3100

Fax: +82-55-772-1909

E-mail address: sgchoi@gnu.ac.kr

Reviewer 2 Report

The manuscript studied the effect of vacuum levels on the properties of beef sirloin. The study is well-designed and presented, which reflects the expertise of the authors. The topic is relevant to the development of food industry, focused on the sous-vide method. The sous-vide method is usually conducted under atmospheric pressure. The study provided an alternative to maintain the texture, color, and oxidative status of meat. Hence, I suggest a minor revision. Here are some points to improve the manuscript:

Abstract: the authors should provide some statistical description on the abstract to support its conclusion.

Line 58: please give the full name of TBA.

Figure 1: I suggest the authors put the explanation of 1-7 in the corresponding part of the figure.

Table 2: How is the unit of the other parameters?

Author Response

Author's Reply to the Review

Dear Editor

We would like to thank you for your consideration and suggestions on our manuscript ‘foods-2233829’, entitled ‘Effect of heating under different vacuum levels on physicochemical and oxidative properties of beef sirloin’. We found the comments helpful and believe our revised the manuscript represents a significant improvement over our initial submission. We have reviewed the comments of the reviewers and have accordingly revised manuscript. In the following pages there are our point-by-point responses to each of the comments of the reviewers.

Reviewers’ comments

Reviewer  #2

The manuscript studied the effect of vacuum levels on the properties of beef sirloin. The study is well-designed and presented, which reflects the expertise of the authors. The topic is relevant to the development of food industry, focused on the sous-vide method. The sous-vide method is usually conducted under atmospheric pressure. The study provided an alternative to maintain the texture, color, and oxidative status of meat. Hence, I suggest a minor revision. Here are some points to improve the manuscript:

[1]

Comment: Abstract: the authors should provide some statistical description on the abstract to support its conclusion. Line 58: please give the full name of TBA.

Response: As a reviewer comments, we have revised (L16 and 70-71)

[2]

Comment: Figure 1: I suggest the authors put the explanation of 1-7 in the corresponding part of the figure.

Response: We already put the explanation in Figure 1 caption (L104-105). Please let us know if there's anything else we can add.

[3]

Comment: Table 2: How is the unit of the other parameters?

Response: Springiness, Chewiness, Gumminess, and Cohesiveness have no unit according to the equation below.

Figure. Texture profile analysis calculation.

  • Hardness (g) = F1
  • Cohesiveness = (d+e)/(a+b)
  • Springiness = (Time 2/Time 1)
  • Gumminess = Hardness x Cohesiveness
  • Chewiness = Gumminess x Springiness

In addition, we have added several references (L435-438 and 472-473).

We hope you will consider this paper as suitable for publication in your journal.

Best regards.

*Corresponding author: Sung-Gil Choi, Professor

Division of Food Science and Technology (Institute of Agriculture and Life Sciences), Gyeongsang National University, Jinju 52828, Korea

phone : +82-10-7143-3100

Fax: +82-55-772-1909

E-mail address: sgchoi@gnu.ac.kr

Reviewer 3 Report

Foods

foods-2233829

Effect of heating under different vacuum levels on physicochemical and oxidative properties of beef sirloin

Dear Editor,

The article deals with the comparison of the physicochemical and oxidative properties of beef sirloin slices heated under atmospheric (101.33 kPa, control) and different vacuum (50.8 kPa and 7.2 Pa) conditions. The topic is good. The manuscript has been generally well designed and written. However, discussion sections should be improved. My specific comments and questions are below;

-       Line 25: some meat products such as sushi, pastrami etc. can be eaten without cooking!

-       How did the researchers keep the meat's internal temperature constant at 75C? Was it based on temperature and time during cooking or internal temperature? Please check!

-       pH: how many g of sample was used?

-       Please give g values not rpm for centrifuge processes!

-       Give the statistical model for your study!

-       Did not the researchers determine moisture content of their samples?

-       Line 178: What could be the possible reason for this consistent?

-       Lines 184 and 185: What could be the possible reason for this?

-       Table 1 and its caption were shifted! Shape it correctly!

-       Lines 251 and 252: What could be the possible reason for this?

-       Table 4: Texture parameters?? Please give and discuss some major saturared (palmitic, stearic), monounsatured (oleic) and polyunsatured fatty acids (linoleic and linolenic acid) individually?

-       Discussion sections should be improved!

Author Response

Author's Reply to the Review

Dear Editor

We would like to thank you for your consideration and suggestions on our manuscript ‘foods-2233829’, entitled ‘Effect of heating under different vacuum levels on physicochemical and oxidative properties of beef sirloin’. We found the comments helpful and believe our revised the manuscript represents a significant improvement over our initial submission. We have reviewed the comments of the reviewers and have accordingly revised manuscript. In the following pages there are our point-by-point responses to each of the comments of the reviewers.

Reviewers’ comments

Reviewer  #3

The article deals with the comparison of the physicochemical and oxidative properties of beef sirloin slices heated under atmospheric (101.33 kPa, control) and different vacuum (50.8 kPa and 7.2 Pa) conditions. The topic is good. The manuscript has been generally well designed and written. However, discussion sections should be improved. My specific comments and questions are below;

[1]

Comment: Line 25: some meat products such as sushi, pastrami etc. can be eaten without cooking!

Response: As a reviewer comments, we have revised (L28).

[2]

Comment: How did the researchers keep the meat's internal temperature constant at 75C? Was it based on temperature and time during cooking or internal temperature? Please check!

Response: A temperature change cannot measured, but a core temperature was only measured after heating at 95°C for 30 min because of the use of an airtight vacuum container.

[3]

Comment: pH: how many g of sample was used?

Response: As a reviewer comments, we have added (L107).

[4]

Comment: Please give g values not rpm for centrifuge processes!

Response: As a reviewer comments, we already give g value for centrifuge process (L137).

[5]

Comment: Give the statistical model for your study!

Response: We already put the explanation ‘2.12. Statistical analysis’ of Material and Methods (L174-178). Please let us know if there's anything else we can add.

[5]

Comment: Did not the researchers determine moisture content of their samples?

Response: We did not measure moisture content because we think the result of cooking loss is enough to judge the weight change.

[6]

Comment: Line 178: What could be the possible reason for this consistent?

Response: The corresponding discussion was mentioned (L199-201).

[7]

Comment: Lines 184 and 185: What could be the possible reason for this?

Response:  The corresponding discussion was mentioned (L225-254).

[8]

Comment: Table 1 and its caption were shifted! Shape it correctly!

Response: As a reviewer comments, we have shifted (L202-205).

[9]

Comment: Lines 251 and 252: What could be the possible reason for this?

Response:  The L*, a*, and b* values are commonly used to measure the color of meat, where L* represents lightness, a* represents redness, and b* represents yellowness. The changes in these color values during cooking can provide important information about the degree of doneness and the overall quality of the meat.

  • L* value: The L* value of meat decreases as it is cooked, due to the Maillard reaction and the loss of water from the meat. This results in a darker color as the meat becomes more cooked.
  • a* value: The a* value of meat increases initially as the meat is cooked, due to the denaturation of myoglobin and the formation of metmyoglobin, which has a brownish-red color. However, as the meat continues to cook and myoglobin breaks down, the a* value decreases, resulting in a less red color.
  • b* value: The b* value of meat also increases initially as the meat is cooked, due to the Maillard reaction and the formation of yellow-brown pigments. However, as the meat becomes more cooked, the b* value decreases, resulting in a less yellow color.

[10]

Comment: Table 4: Texture parameters?? Please give and discuss some major saturared (palmitic, stearic), monounsatured (oleic) and polyunsatured fatty acids (linoleic and linolenic acid) individually?

Response: As a reviewer comments, we have revised in Table 4 and added (L361-363).

[10]

Comment: Discussion sections should be improved!

Response: As commented, we have improved (L209-216, 235-236, 295-297, 336-337, 342-344, and 361-363) in '3. Results and Discussion’.

In addition, we have added several references (L435-438 and 472-473).

We hope you will consider this paper as suitable for publication in your journal.

Best regards.

*Corresponding author: Sung-Gil Choi, Professor

Division of Food Science and Technology (Institute of Agriculture and Life Sciences), Gyeongsang National University, Jinju 52828, Korea

phone : +82-10-7143-3100

Fax: +82-55-772-1909

E-mail address: sgchoi@gnu.ac.kr

Round 2

Reviewer 1 Report

There are some minor points asking author to reply as follow:

 - Why determine only 2 level of vacuum (50% and 99.9%) since the 50% is too least while the 99.9% is difficult to use in the real situation/industry. The comparison between half and extreme vacuum levels on oxidations and others was pretty know somehow.

- Author mentioned that the sample used a specially designed airtight vacuum container which is completely isolated from the external environment, not vacuumized polypropylene pouch. How this information were applied in the real situation? Or how’s different among this method and traditional pounch?

Author Response

Comments :

- Why determine only 2 level of vacuum (50% and 99.9%) since the 50% is too least while the 99.9% is difficult to use in the real situation/industry. The comparison between half and extreme vacuum levels on oxidations and others was pretty know somehow.

- Author mentioned that the sample used a specially designed airtight vacuum container which is completely isolated from the external environment, not vacuumized polypropylene pouch. How this information were applied in the real situation? Or how’s different among this method and traditional pounch?

Response :

  We consider that this study has not only an academic significance but also is valuable as basic data in meat industry. The reason why the heat treatment at 2 levels of vacuum (50% and 99.9%) using an airtight vacuum container is valuable not only to clearly identify the effect of internal vacuum condition and thermal treatment on physicochemical and oxidative properties, but also to investigate dependence on the properties without influence from other factors. Especially, as a new finding showed that there was little browning under oxygen-deficient condition, so it can be used as a basic data for myglobin change/denaturation research during cooking. In addition, the container can be used when developing a vacuum controlled sousvide recipe.